# MDEval: Evaluating and Enhancing Markdown Awareness in Large Language Models

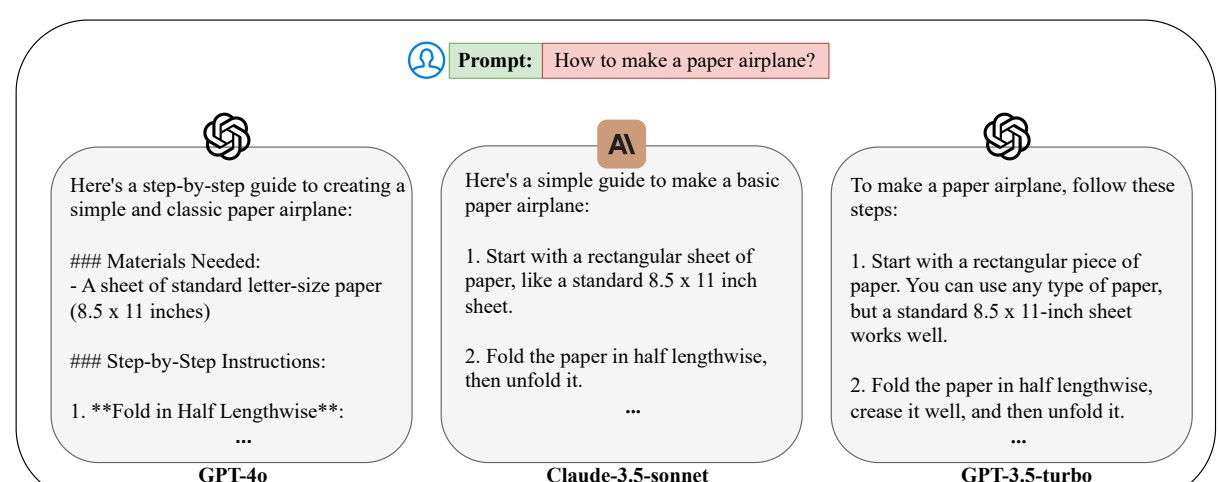

**Figure 1: Differences of `Markdown Awareness` of LLMs under the same input prompt. GPT-4o is shown to have superior capability in generating outputs with Markdown format. Although Calude-3.5-sonnet is generally known to be more advanced than GPT-3.5-turbo, they demonstrate comparable performance with respect to this particular metric.**

## Abstract

Large language models (LLMs) are expected to offer structured Markdown responses for the sake of readability in web chatbots (e.g., ChatGPT). Although there are a myriad of metrics to evaluate LLMs, they fail to evaluate the readability from the view of output content structure. To this end, we focus on an overlooked yet important metric — `Markdown Awareness`, which directly impacts the readability and structure of the content generated by these language models. In this paper, we introduce MDEval, a comprehensive benchmark to assess `Markdown Awareness` for LLMs, by constructing a dataset with 20K instances covering 10 subjects in English and Chinese. Unlike traditional model-based evaluations, MDEval provides excellent interpretability by combining model-based generation tasks and statistical methods. Our results demonstrate that MDEval achieves a Spearman correlation of 0.791 and an accuracy of 84.1% with human, outperforming existing methods by a large margin. Extensive experimental results also show that through fine-tuning over our proposed dataset, less performant open-source models are able to achieve comparable performance to GPT-4o in terms of `Markdown Awareness`. To ensure reproducibility and transparency, MDEval is open sourced at https://anonymous.4open.science/r/MDEval-Benchmark-1730/.

## CCS Concepts

• **Information systems → Information systems applications**; *Web data description languages.*

## Keywords

Large Language Models, Benchmark, Markdown Awareness, Structured Response, Web Chatbot Readability

**ACM Reference Format:**
Anonymous Author(s). 2025. MDEval: Evaluating and Enhancing Markdown Awareness in Large Language Models. In *Proceedings of The Web Conference (WWW 2025)*. ACM, New York, NY, USA, 11 pages. https://doi.org/XXXXXXX.XXXXXXX

## 1 Introduction

Recently, the rapid progress of Large Language Models (LLMs) has considerably reshaped the information industry [34]. One of the phenomenal applications is web chatbots [33], such as ChatGPT and Claude, which are able to generate reasonable and thoughtful responses to prompts from various domains. Users can thus leverage LLMs to boost their productivity significantly, including information retrieval, ideation and brainstorming, and task planning [35]. When it comes to a prompt that is worthy to generate an information-rich response, users would expect that the output should be well-structured for better readability [2, 29] and visual

effect on web. As a result, modern web chatbots powered by advanced LLMs (e.g., GPT-4o) tend to output responses with Markdown format, which is web friendly owing to several outstanding Markdown parsers (e.g., `marked.js`) for HTML. As illustrated in Figure 1, GPT-4o is shown to have superior capability in outputting text with structured Markdown even without explicit instructions, and it makes use of headings, bolding, and listings, etc., to increase the readability of a long response. Although Calude-3.5-sonnet is generally known to be more advanced than GPT-3.5-turbo, they demonstrate comparable performance with respect to this particular metric. As a result, a natural question arises: **How good are LLMs in terms of Markdown output capability?** To this end, in this paper, we introduce a novel LLM metric, termed Markdown Awareness, and then propose a comprehensive benchmark to evaluate LLMs' Markdown output capability without explicit instructions in a zero-shot setting.

Evaluation is a fundamental task for LLMs [23]. Due to the diversity and complexity of tasks in real world, an abundance of evaluation benchmark has been proposed to focus on one or several tasks, including coding [5], math [9], and multi-turn questions [19]. A large variety of evaluation metrics, covering answer relevancy, faithfulness, contextual recall, contextual precision, hallucination, toxicity, and bias [1], are introduced to fit different requirements in practical applications. However, to the best of our knowledge, there is no benchmark evaluating Markdown Awareness for LLMs. We argue that an LLM with high Markdown Awareness has considerable impact on numerous web applications, such as chatbots, in terms of readability. For example, ### and **...** are rendered as headings and bold texts on web pages in Figure 1, respectively, so an output with a higher Markdown Awareness score offers better visual effect, presenting a clear structure, and emphasized key points. Thus, Markdown Awareness helps to reduce cognitive load, making it easier for readers to process and understand the material [15, 24]. Notably, Markdown Awareness is quite significant for outputs in the field of science, technology, engineering, and mathematics, in which standard code and math elements[1] are expected for better rendering on web pages. As illustrated in Figure 2, even advanced LLMs like Claude-3.5-sonnet exhibit a significant gap in their ability to output mathematical equations.

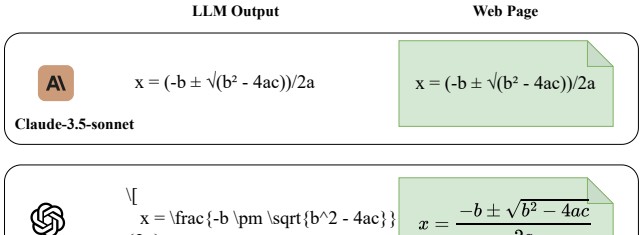

**Figure 2: GPT-4o-mini is able to output mathematical equations in extended Markdown with LaTeX support, while Claude-3.5-sonnet only generates plain text.**

---

[1]Although both inline math and block math are not parts of Markdown, they are supported as a de-facto feature in extended Markdown and adopted by many websites owning to the prestigious library `katex`.

However, it is non-trivial to design a benchmark to evaluate the Markdown Awareness for LLMs due to three major challenges:

- *(C1) Insufficient Datasets.* Dataset is a foundation for benchmark evaluation [20, 36]. Although it is feasible to generate a dataset of Q&A pairs from either models or humans, it is difficult to determine a single "expected" output (i.e., answer) per question (i.e., prompt) because we mainly focus on the style and structure for the sake of better readability, rather than the content of LLMs' responses[2].

- *(C2) Metric Validity.* Developing a reasonable metric for Markdown quality is daunting. Traditional statistical scoring methods (e.g., BLEU [27]) heavily depend on ground-truth (e.g., expected) outputs, which are difficult to be established due to the limitation of datasets mentioned above. On the other hand, pure model-based methods (e.g., G-Eval [21]), evaluating LLMs' outputs through LLMs, are neither stable nor explainable.

- *(C3) Metric Quantity.* Metrics, such as BLEU and ROUGE [7], are *content-oriented*, while Markdown Awareness, in fact, is *structure-oriented*. A straightforward method is firstly to count the Markdown element's frequency, and assign a weight to each Markdown element. Then a weighted sum is computed as the score of Markdown Awareness. However, weights are hard to be determined in practice, and the sum cannot be easily normalized.

To tackle the issues above, we propose MDEval, a comprehensive benchmark to evaluate the Markdown quality of LLMs' outputs. The key techniques of MDEval include:

*(1) A Ground-truth-free Dataset (C1).* Existing benchmarks always assume that there is a single ground-truth. We acknowledge the fact that LLMs show differing capabilities (e.g., answer relevancy, and hallucination) given the same prompt, so when it comes to the structure of the response, the expected output should be model-dependent. Thus, we construct a ground-truth-free dataset with 20K instances, covering 10 subjects.

*(2) An Intermediate Rewrite Phrase (C2).* Since the dataset is ground-truth-free, it seems that the only feasible yet unstable way is to evaluate LLMs' outputs through black-boxed LLMs. Instead, we improve this "LLMs as judges" method by introducing another generation task using LLMs. To be specific, we propose a novel rewrite strategy to obtain a model-specific intermediate reference on-the-fly, which can be exploited in the next phrase of statistical evaluation with excellent explainability. Note that the content itself of the rewritten "stylish" counterpart remains (mostly) unchanged.

*(3) A Structure-oriented Metric (C3).* After the intermediate rewrite phrase, we are able to compute the evaluation score through comparison. Note that in Markdown, multiple symbols can represent the same semantic meaning. For example, both - and * indicate unordered lists. In addition, extracting Markdown elements would lose the structure information from the text itself, because the plain text can also be considered being *weak* structured. To this end, we propose a novel structure-oriented metric by computing the

---

[2]Textual content itself definitely has substantial influence on readability, but it is beyond the scope of this paper.

 

edit distance between the original output and the model-specific reference after transforming the response into HTML format[3].

The contribution of our work can be summarized as follows:

- To the best of our knowledge, MDEval is the first benchmark to evaluate the quality of LLMs' Markdown output, and we proposed a novel structure-oriented metric, named `Markdown Awareness`, which is important yet overlooked in web chatbots.
- MDEval provides a dataset with 20K instances covering 10 subjects in Chinese and English, in which every instance is worthy of a well-formatted response. And we reported `Markdown Awareness` performance across 9 mainstream LLMs based on the dataset.
- The proposed `Markdown Awareness` is of validity by combining model-based generation tasks and statistical methods seamlessly in an evaluation pipeline, and we also showed that it is able to align with humans' preference.
- We demonstrated that through fine-tuning over the dataset constructed above, less performant open-source models can achieve comparable performance to GPT-4o in terms of `Markdown Awareness`.

## 2 Related Work

### 2.1 Evaluation Tasks

In recent years, LLMs have made significant progress in the field of natural language processing, making the evaluation of their performance a central task in research [6, 26]. The scope of evaluation tasks is broad, encompassing areas such as natural language understanding [25], text generation [14], and instruction following [39]. To effectively evaluate the performance of these models, researchers have designed multiple benchmarks. For example, GLUE [31] focuses on word-sentence level language understanding, dialogue understanding, information retrieval and answering, language generation, and mathematical reasoning. HELM [4] concentrates on question answering, information retrieval, and toxicity detection. Our work is closely related to Chatbot Arena developed in [6] which provides valuable insights into general chatbot performance based on crowdsourcing, but it does not integrate Markdown quality into the evaluation framework.

On the other hand, some benchmarks focus on specific types of tasks. For instance, NL2Code [37] primarily targets code generation, assessing models' capabilities in programming language understanding and generation. ARC [8] tests LLMs' performance on elementary science questions. These task-specific benchmarks can reveal models' strengths and weaknesses more precisely within a certain domain. However, to the best of our knowledge, none of the existing work pays attention to `Markdown Awareness`, which plays a key role in web chatbots for better readability.

### 2.2 Evaluation Metrics

The diversity of evaluation metrics is closely related to the complexity of evaluation tasks. Generally speaking, there are two lines of evaluation metrics for text content.

- **Statistical Metrics**. Traditional metrics based on text comparison or purely on text generation, such as BLEU [27], ROUGE [7], and METEOR [3], which are essentially word-based by measuring the overlap between generated text and reference text. Clearly, statistical metrics suffer from limited reasoning capabilities because they fail to take the semantics into account.
- **Model-based Metrics**. Metrics, which rely only on an LLM, gain much attention recently due to the increasing capability of LLMs. For example, BLEURT [28], a BERT-based scorer, solves the issue of poor correlation with human judgments. On the other hand, metrics like Word Mover's Distance [18], and BERTScore [38] evaluate the quality of generated content based on model-based similarity. Recently, metrics like G-Eval [21] aim to provide better human alignment using chain-of-thoughts. However, these approaches face considerable computational and financial costs due to their reliance on majority-voting strategies with a leading LLM.

In the meanwhile, several metrics (e.g., GPTScore [13], and Self-CheckGPT [22]) adopt a hybrid way to make a balance between reliability and accuracy by combining two kinds of metrics above. Despite the abundance of evaluation metrics, none adequately assess a model's `Markdown Awareness`.

### 2.3 Web Content Readability

Web content readability [11] is a well-studied topic since the World Wide Web (WWW) has gained a worldwide popularity in 1990. As a result, a large body of work is dedicated to evaluating the readability of web content. For example, Kanungo et al. [17] adopt the gradient boosted decision tree to predict the readability of web search summaries. The fuzzy logic values based method was developed in [16] for a better approximate confinement of partial correspondence. The authors in [29] focused on how the table element affects readability on mobile devices.

However, existing work ignores the web content readability of chatbots from the perspective of Markdown quality, and we believe this underestimated feature is significant for language models. There remains a lack of professional benchmarks for evaluating Markdown readability of web content.

## 3 Proposed Method

In this section, we firstly provide the big-picture of MDEval, which is a pipeline evaluation, consisting of model-generating, response rewriting, structure extracting and scoring.

As illustrated in Figure 3, given a specific task (e.g, "How to make a paper airplane") and a target LLM (e.g., Llama [10]), the initial output generated by the target LLM may lack proper Markdown formatting (Phrase 1). To this end, MDEval subsequently invokes an advanced LLM (e.g., GPT-4o) to rewrite the initial response to generate a well-structured Markdown counterpart while preserving the original textual (Phrase 2). The rewritten response serves as a model-dependent reference because it should be constructed on-the-fly, rather than from a pre-collected dataset. Since `Markdown Awareness` is structure-oriented, MDEval further converts the two responses above into HTML contents and then extracts the HTML

---

[3]To handle extended math elements (e.g., \[ \]), we introduce custom HTML tags for them, and it is detailed in Section 3.

tags, respectively (Phrase 3). Finally, the edit distance is computed between the extracted HTML tags, and a lower edit distance often indicates a superior Markdown Awareness (Phrase 4). In what follows, we will provide a comprehensive analysis of each phrase in MDEval, and an evaluation system which we developed from scratch in order to test the human alignment.

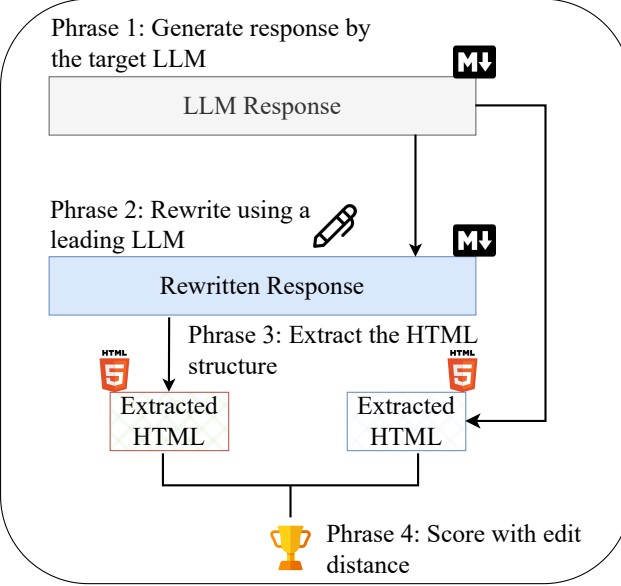

**Figure 3: The overall framework of MDEval. Given the task and a target LLM, it firstly generates the response to the prompt (Phrase 1), and then MDEval leverages an advanced LLM (e.g., GPT-4o) to rewrite the response as a model-dependent reference (Phrase 2). Next, MDEval extracts the HTML structures from two responses (Phrase 3). Finally, the score is computed based on the edit distance (Phrase 4).**

## 3.1 Detailed Methodology Phrases

Let $\mathcal{T}$ be the set of all tasks (i.e. prompts) in the dataset, and $\mathcal{L}$ be the set of all LLMs in MDEval. Phrase 1 can be defined as $f : \mathcal{T} \times \mathcal{L} \rightarrow \mathcal{R}$, where $\mathcal{R}$ is the set of generated responses. The task in $\mathcal{T}$ is expected to be answered with a well-structured response, and in the most applications (e.g., web chatbots), it should be in Markdown format to highlight key points, provide code snippet, and create the lists, etc. Note that the prompt should not include any explicit instruction to let LLMs generate a Markdown response, because conversations between humans and AI should be question-oriented and natural, without excessive instructions. Otherwise, this task would degrade to an example of instruction following.

As for Phrase 2, a rewriting procedure is conducted via prompt engineering. The carefully crafted prompt $\hat{\mathcal{P}}$ is to instruct a leading LLM $\hat{\mathcal{L}}$ (e.g., GPT-4o) to rewrite a response in $\mathcal{R}$ using the Markdown format. So Phrase 2 can be defined as $g : \hat{\mathcal{P}} \times \hat{\mathcal{L}} \times \mathcal{R} \rightarrow \hat{\mathcal{R}}$. To ensure fairness in the evaluation, both $\hat{\mathcal{P}}$ and $\hat{\mathcal{L}}$ are singleton, and we omit those notations if the context is clear. The prompt can be:

```
Given the text below, rewrite it using
Markdown format to make the output more
structured, and increase the readability.

Note that if possible, keep the content the
same, just adjust the formatting.
###
{text}
```

Because Markdown Awareness is structured oriented and the textural content itself could potentially introduce bias into the evaluation process, we further extract the structured elements from both $\mathcal{R}$ and $\hat{\mathcal{R}}$ in Phrase 3. A straightforward way is to find Markdown syntax elements through regular expressions, but it can be error-prone due to various Markdown flavors. As an alternative strategy, we propose an effective method by first converting the Markdown content into HTML format, followed by the extraction of HTML tags. It is noteworthy that mathematical elements powered by LATEX cannot be properly converted through this process. To address this limitation, we introduce a custom `<math>` tag specifically designed to accommodate these elements. In this paper, we denote this process as *HTMLify*.

To ensure explainability of MDEval, we adopt a statistical way to compute the score as Markdown Awareness in Phrase 4. To be specific, given a task $t \in \mathcal{T}$, consider an HTMLify response $r$ generated by a language model $l \in \mathcal{L}$, and its corresponding rewritten HTMLify response $\hat{r}$, Markdown Awareness $MA(t, l)$ is defined as the normalized edit distance (i.e., Levenshtein distance), ranging from 0 to 1:

$$MA(t, l) = 1 - \frac{editDistance(r, \hat{r})}{\max(len(r), len(\hat{r}))}, \quad (1)$$

where $editDistance(\cdot, \cdot)$ is measured by counting the minimum number of operations required to transform $r$ into $\hat{r}$ (or vice versa), and $len(\cdot)$ is the length of a given string. Generally, a higher Markdown Awareness of a response generated by an LLM implies a better Markdown structure. For example, given an LLM with great Markdown output ability, the edit distance between $r$ and $\hat{r}$ is very small, and thus its Markdown Awareness approximates 1.

## 3.2 Human Alignment Evaluation System

Unlike traditional tasks such as text summarization where human ratings are available [12], Markdown Awareness presents considerable challenges in evaluation without human preference datasets. To this end, we build a human alignment evaluation system for MDEval, and a test system can be publicly visited at https://md-eval-human.pages.dev. Since it is difficult for a human to compare or rate the responses generated by a dozen of LLMs directly (in this paper, the size of $\mathcal{L}$ is 9), we adopt a pairwise comparison approach inspired by Chatbot Arena [6].

**Evaluation Interface.** A uniformly sampled task $t \in \mathcal{T}$ and its two random corresponding responses generated by anonymous $\mathcal{L}_i$ and $\mathcal{L}_j$ ($i \neq j$), respectively, are rendered in the web pages, and human experts are asked to choose their preferences. Furthermore, to enhance reliability and user flexibility, the interface incorporates a toggle button that allows users to switch between viewing the

raw Markdown source code and the rendered content. As a result, a comparative dataset $\mathcal{A} = \{(t, \mathcal{L}_i, \mathcal{L}_j, h)\}$ can be gathered through crowdsourcing, where $h \in \{W, L, T\}$. Specifically,

(1) If $h = W$, it indicates that a human expert prefers $\mathcal{L}_i$ over $\mathcal{L}_j$.
(2) Conversely, when $h = L$, it indicates that $\mathcal{L}_j$ is preferred.
(3) Otherwise (i.e., $h = T$), it results in a tie between two responses.

**Rankings Through Elo Ratings.** Constrained by the law of large numbers, the LLMs rankings based on winning rates are rarely applicable due to the unbalance between human labors and possible samplings. Instead, the Elo rating methodology [32] makes it possible to infer stable expected scores from wins, losses, and draws. Let $S_i$ and $S_j$ be the rating of $\mathcal{L}_i$ and $\mathcal{L}_j$, respectively, and for an LLM $\mathcal{L}_i$, its probability of winning plus half its probability of drawing is given by

$$P(\mathcal{L}_i) = \frac{Q_i}{Q_i + Q_j}, \qquad (2)$$

where $Q_i = 10^{S_i/d}$, and $Q_j = 10^{S_j/s}$. Specifically, $d$ is a number which represents a significant difference in Markdown Awareness between two LLMs. The probability above is also known as the *expected score*. And similarly, the expected score of $\mathcal{L}_j$ is

$$P(\mathcal{L}_j) = \frac{Q_j}{Q_i + Q_j}. \qquad (3)$$

Without loss of generality, only $\mathcal{L}_i$ is discussed in the following. The score is computed in an iterative way. Initially, every LLM is assigned a base rating. For each instance $(\mathcal{L}_i, \mathcal{L}_j, h) \in \mathcal{A}$, we can assign an actual score $P(\mathcal{L}_i^*)$ for $\mathcal{L}_i$ based on $h$. In this paper, the detailed strategy is:

(1) If $h = W$, $P(\mathcal{L}_i^*) = 1$.
(2) If $h = L$, $P(\mathcal{L}_i^*) = 0$.
(3) Otherwise (i.e., $h = T$), $P(\mathcal{L}_i^*) = 0.5$.

The iterative updating rating is formulated by introducing a K-factor:

$$S_i' = S_i + K \times (P(\mathcal{L}_i^*) - P(\mathcal{L}_i)). \qquad (4)$$

To build a reasonable $\mathcal{A}$, it is essential to choose the appropriate parameters, including $d$ and $K$, to reflect the sensitivity of ratings for a high accuracy, while these parameters are highly related with the ratio of ties. Upon analysis of the experimental results, as illustrated in Figure 4, we observe that the ratio of ties approximates the result reported in [6]. Consequently, we adopt the similar settings with $d = 400$ and $K = 10$ in MDEval.

**Human Alignment Test.** To ensure the validity of the proposed metric, MDEval conducts comprehensive human alignment tests based on $\mathcal{A}$. Firstly, the *record-level* accuracy is evaluated by checking whether the score ordering of MDEval matches with the comparative tuple in $\mathcal{A}$, and it is formulated by

$$\frac{\sum\limits_{(t, \mathcal{L}_i, \mathcal{L}_j, h) \in \mathcal{A}} ind(t, \mathcal{L}_i, \mathcal{L}_j, h)}{|\mathcal{A}|}, \qquad (5)$$

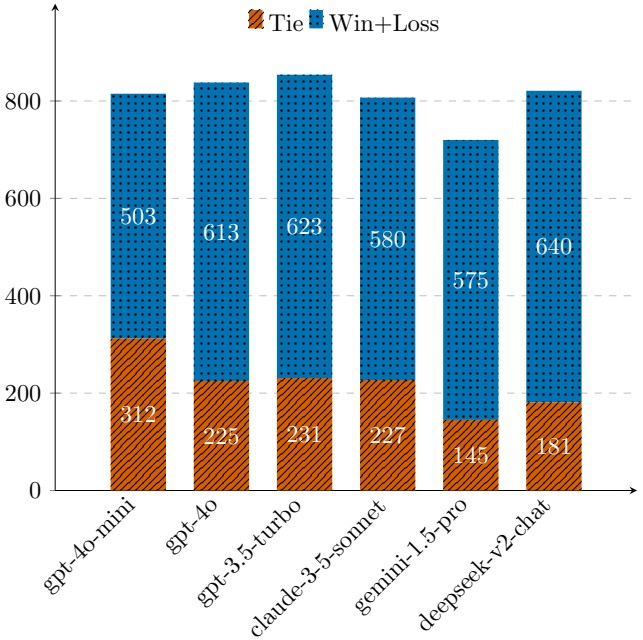

**Figure 4: The count of ties vs. wins + losses at an early snapshot of $\mathcal{A}$ in our system. We can observe that the ratio of ties ranges from 20% to 38%, and it stays relatively constant as the amount of collected data grows.**

where $ind(\cdot, \cdot, \cdot, \cdot)$ is an indicator function which can be defined by the following equation, where $\mathbb{1}(\cdot) = 1$ when its parameter holds true, and $\mathbb{1}(\cdot) = 0$ otherwise:

$$\begin{aligned} \mathbb{1}((MA(t, \mathcal{L}_i) > MA(t, \mathcal{L}_j) \wedge h = W) \vee \\ (MA(t, \mathcal{L}_i) < MA(t, \mathcal{L}_j) \wedge h = L) \vee \\ (MA(t, \mathcal{L}_i) = MA(t, \mathcal{L}_j) \wedge h = T)) \end{aligned} \qquad (6)$$

Secondly, the *task-level* correlation coefficients (e.g., Spearman and Kendall) [21] are evaluated to test whether the proposed Markdown Awareness aligns with human preferences based on Elo ratings.

## 4 The MDEval Dataset

In this section, we present an analysis of the MDEval dataset which we have developed from scratch. We begin by discussing the approach employed in its construction, providing a detailed account of the data collection and curation processes. Subsequently, we offer insights of how to fine tune an LLM utilizing this dataset, with a focus on enhancing its Markdown Awareness.

### 4.1 Description of the Dataset

To the best of our knowledge, this is the first dataset aiming to evaluate the Markdown quality of LLMs. As discussed in Section 1, the dataset consists of tasks (prompts) without established ground truths, offering dual advantages in terms of time and financial efficiency. The majority of the dataset is built in a hybrid way by asking ChatGPT to generate questions based on the raw texts from Wikipedia. Since Markdown Awareness makes sense only

when the expected output is well-structured, we further conduct a post-processing to feed the tasks into a local-deployed LLM with chain-of-thoughts to filter unqualified prompts which either contain explicit style/format instructions, or tend to generate short responses where the Markdown outputs are not necessary. The total size of the MDEval dataset is 20K instances in both English (70%) and Chinese (30%). To enhance its diversity among various domains, the MDEval dataset covers 10 subjects, in which the proportion of each subject is about 10%, including 1) Business and Economics; 2) Social Sciences and Human Rights; 3) Environment and Sustainability; 4) Science and Technology; 5) Law, Legal Studies and International Relations; 6) History, Geography and Cultural Studies; 7) Education and Learning (Math, Programming, etc.); 8) Health, Wellness and Fitness; 9) Morals and Ethics; and 10) Psychology and Behavioral Sciences. Additionally, the winning matrix generated in our human alignment evaluation system is a valuable resource for post-training.

## 4.2 Fine-tuning with the Dataset

The preprocessing of training data for LLMs often involves the removal of Markdown elements, and it is a great practice that has been widely adopted in the field of natural language processing. This approach is exemplified in the development of Llama 3, as documented by Meta [10]. As a result, `Markdown Awareness`, in fact, is enhanced during the fine-tuning stage.

Inspired by the idea of "quality is all you need" [30], we propose a carefully curated dataset to enhance an LLM's `Markdown Awareness` by supervised fine-tuning (SFT). To be specific, given a task, its ground truth output is the response generated by the most advanced LLM reported in our human alignment evaluation system. This innovative SFT dataset also serves as a significant reference point for evaluating and enhancing LLMs' proficiency in Markdown generation. As shown in Section 5, less performant open-source models are able to achieve comparable performance to GPT-4o in terms of `Markdown Awareness` through fine-tuning.

## 5 Experiments

### 5.1 Experimental Setting

In this work, we use the MDEval dataset which is developed for `Markdown Awareness` evaluation. As discussed in Section 4.1, it contains 20K instances (i.e., $\mathcal{T}$) in both English and Chinese, covering the following 10 roughly equally-sized subjects. We consider 9 widely used LLMs (i.e., $\mathcal{L}$) in the benchmark, spanning from state-of-the-art large to small models, and including both proprietary and open-source implementations[4]: gpt-4o-mini, gpt-4o, gpt-4-turbo, gpt-3.5-turbo, calude-3.5-sonnet, gemini-1.5-pro, deepseek-v2-chat, baichuan2-13b-chat-v1, and llama-3.1-8b. It is worth noting that MDEval exhibits significant extensibility, facilitating the trivial integration of more instances into the dataset and the incorporation of other LLMs into the benchmark framework directly.

Although `Markdown Awareness` has not been studied yet, several baselines can be developed to evaluate the Markdown quality of LLMs' output. Based on the dataset, we compare our MDEval with three methods as shown below:

---

[4]In our experiments, gpt-4o-mini is gpt-4o-mini-2024-07-18; gpt-4-turbo is gpt-4-turbo-2024-04-09; and claude-3-5-sonnet is claude-3-5-sonnet-20240620.

(1) **Pure LLM-based (P-LLM)**. This method purely relies on NLP models to provide a score by prompt engineering. In this paper, we use GPT-4o as the judge. To avoid parsing issues caused by LLMs hallucinations, we use structured outputs[5] which conform to a pre-defined schema with the predicted score.

(2) **Referenced LLM-based (R-LLM)**. Due to the inherent probabilistic feature, P-LLM can be unreliable. To solve this issue, a reference is generated by a leading model with explicit Markdown hints. Subsequently, R-LLM feeds GPT-4o the reference as the context by leveraging an adjusted prompt, and asks GPT-4o to output the final predicted score with the reference in mind.

(3) **Decayed Rule-based (D-Rule)**. This method is essentially a statistical scorer through empirically derived heuristic rules. To obtain a normalized score, a reference is also required, which is generated utilizing methodology analogous to that employed in the R-LLM. A straightforward solution is to count the number of Markdown enumerate, and (optional) assign weights to different Markdown elements. However, this approach is susceptible to a bias favoring verbose responses. To mitigate this limitation, we propose a penalized strategy that incorporates a decayed factor, thereby optimizing both content quality and concision in terms of `Markdown Awareness`. Due to the limit of space, D-Rule is elaborated in Appendix A.1.

To verify the effectiveness of the proposed `Markdown Awareness` in MDEval, we mainly focus on two types of human alignment test, including the *record-level* accuracy and the *task-level* correlation, as detailed in Section 3.2.

In this paper, we would like to answer the following significant research questions through comprehensive experimental evaluations:

- **RQ1:** How good are the LLMs competitors in terms of `Markdown Awareness`?
- **RQ2:** How good is MDEval in terms of human alignment compared to other methods?
- **RQ3:** Is `Markdown Awareness` correlated with LLMs' other capabilities in the public leaderboard?
- **RQ4:** How effective is fine-tuning in terms of `Markdown Awareness`?
- **RQ5:** Does `Markdown Awareness` vary across different subjects and languages?

### 5.2 RQ1

To answer RQ1, we compute the average `Markdown Awareness` for every evaluated language model $l \in \mathcal{L}$ based on Equation (1). The score of each evaluated LLM is summarized in Table 1, which is sorted in the descending order of the score. As illustrated in Table 1, deepseek-v2-chat outperforms other LLMs by a large margin, including the state-of-the-art gpt-4o. As of the time of writing, deepseek-v2-chat only ranks 31st, while gpt-4o ranks 1st with respect to the overall performance [6]. On the other hand, llama3.1-8b, a popular open source LLM with only 8 billion parameters which

---

[5]https://platform.openai.com/docs/guides/structured-outputs

can be deployed on a personal computer, beats both calude-3.5-sonnet and gpt-3.5-turbo in MDEval. Therefore, we can observe that there is an obvious gap between overall capabilities and Markdown Awareness. A further question is to investigate the correlation between Markdown Awareness and other capabilities, and this is exactly RQ3 in Section 5.4.

**Table 1: Markdown Awareness Scores and Rankings**

| Ranking | LLM | Score | Average Ranking |
|---------|-----|-------|-----------------|
| #1 | **deepseek-v2-chat** | **0.946** | #1 |
| #2 | gpt-4o | 0.865 | #2 |
| #3 | gpt-4o-mini | 0.830 | #3 |
| #4 | gemini-1.5-pro | 0.803 | #5 |
| #5 | gpt-4-turbo | 0.787 | #4 |
| #6 | llama-3.1-8 | 0.710 | #6 |
| #7 | calude-3.5-sonnet | 0.569 | #7 |
| #8 | gpt-3.5-turbo | 0.482 | #8 |
| #9 | baichuan2-13b-chat-v1 | 0.171 | #9 |

In addition, we also adopt another average ranking-based method to compute the rankings, as shown in the fourth column of Table 1. We can find that the rankings obtained from this alternative method closely align with those derived from our primary approach. This consistency across different ranking methodologies reinforces the robustness of our findings. The strategy of average ranking is like the GPA calculation from letter grades (see more in Appendix A.2).

## 5.3 RQ2

Foremost, we discuss the results of human voting data collected in our human alignment evaluation system, which is the cornerstone to answer RQ2. We can gain important insights from human experts' voting in terms of Markdown Awareness, and the Elo ratings of each LLM by bootstrap method is depicted in Figure 5. The empirical data indicate that the Elo ratings exhibit remarkable stability, with the standard deviation error consistently less than 25 in the majority of instances. Moreover, the human-derived rankings demonstrate a strong alignment with MDEval, as evidenced in Table 1.

Next, we are going to answer RQ2 directly by comparing MDEval with another three methods in terms of *record-level* accuracy and *task-level* correlation, in which ties are skipped. The result is summarized in Table 2, and we can find that the average accuracy and correlation of MDEval outperform LLM-based methods by a large margin. Notably, contrary to our intuitions, feeding a reference to an LLM (i.e., R-LLM) does not bring any benefit for Markdown Awareness. This unexpected finding challenges prevalent assumptions regarding the utility of external references in enhancing LLMs performance. Furthermore, a well-designed rule-based method (i.e. D-Rule) demonstrates superior performance compared to model-based implementations (both P-LLM and R-LLM), while also exhibiting notable cost-effectiveness.

*Discussion.* Due to the inherent probabilistic feature of LLMs, we may adopt majority-voting when using LLM-based methods, and this would cause a considerable overhead. As for D-Rule (see more in Appendix A.1), it relies on less-interpretable heuristic rules

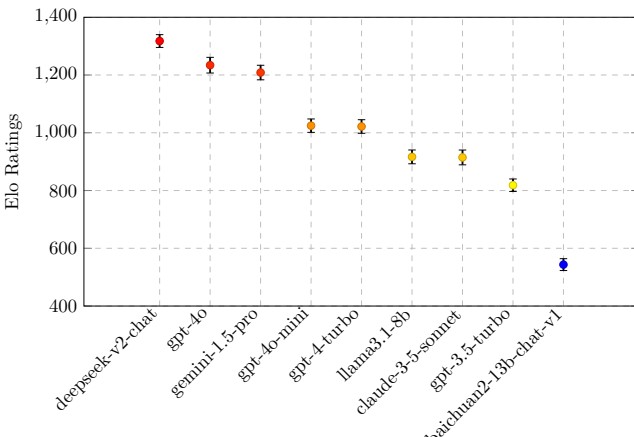

**Figure 5: The Elo ratings and confidence intervals from human evaluation by voting. To derive a more stable result, we use the bootstrap method with 1000 rounds.**

**Table 2: Accuracy and Correlation**

| Method | Accuracy | Spearman | Pearson | Kendall |
|--------|----------|----------|---------|---------|
| MDEval | 84.1% | **0.791** | **0.844** | 0.670 |
| P-LLM | 73.8% | 0.779 | 0.783 | **0.674** |
| R-LLM | 70.7% | 0.710 | 0.705 | 0.600 |
| D-Rule | 83.4% | 0.757 | 0.685 | 0.625 |

which are hard to be standardized. On the other hand, MDEval, as a hybrid scorer combining both statistical-based and model-based methods, offers excellent interpretability without human-crafted rules, and achieves the best overall performance.

## 5.4 RQ3

To answer RQ3, we collect the LLMs rankings from Chatbot Arena in several different dimensions, including Instruction Following, English, Chinese, Math, Coding, Hard Prompt (All), and Longer Query, and then compute the average Spearman correlation. The results are shown in Figure 6. As we can see, Markdown Awareness is closely related with English/Chinese/Coding/Longer Query. Because MDEval still focuses on text information, the ability demonstrated in languages matters. We also find that English shows a higher correlation than Chinese, and this discrepancy may be attributed to a larger proportion of English tasks in our dataset. On the other hand, the coding performance often implies Markdown Awareness as code is structured to some extent. Similarly, a longer query also tends to be more structured.

## 5.5 RQ4

As shown in Table 1, baichuan2-13b-chat-v1 performs the worst among all competitors. To enhance its Markdown Awareness through transfer learning based on the dataset developed in Section 4.2, we adopt the QLoRA method to fine tune baichuan2-13b-chat-v1. As reported in LLaMA-Factory [40], it takes about 20GB GPU memory

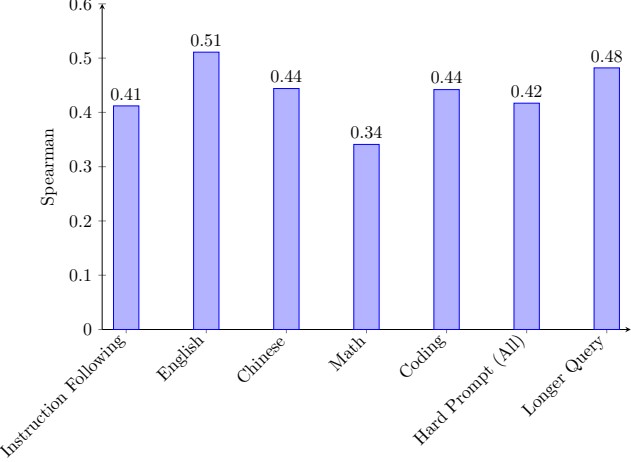

**Figure 6: Spearman correlation between `Markdown Awareness` and LLMs' capabilities in the public leaderboard.**

for a 13B model with 8-bit precision. To answer RQ4, we conduct the experiments on a Linux machine with an Nvidia 4090 GPU. With the increasing of the size of SFT data, the performance of the model exhibits significant improvements, as shown in Figure 7. It reaches 0.73 when the size of instances is 1600, approximating the gpt-4 level.The results verify the effectiveness of fine-turning for `Markdown Awareness` and the high quality of our dataset.

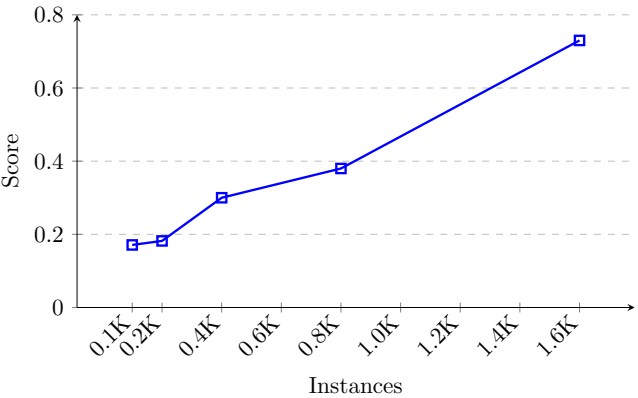

**Figure 7: Model performance in terms of `Markdown Awareness` with the size of SFT instances.**

### 5.6 RQ5

The performance and capabilities of LLMs frequently exhibit heterogeneity across diverse domains and linguistic contexts. In light of this variability, we examine how `Markdown Awareness` performance differs across various subjects and languages.

The average score of `Markdown Awareness` in English and Chinese is reported in Figure 8. Generally speaking, linguistic contexts have little influence on the score in MDEval, except llama3.1-8b and baichuan2-13b-chat-v1. To be specific, llama3.1-8b in English

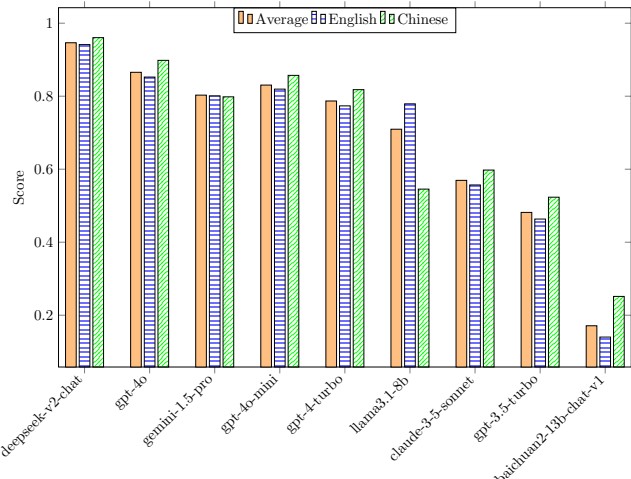

**Figure 8: `Markdown Awareness` in English and Chinese of each LLM. Generally, linguistic contexts have little influence on the score in MDEval.**

context (0.779) performs much better than it in Chinese context (0.545). On the other hand, baichuan2-13b-chat-v1 shows the opposite trend, demonstrating superior performance in Chinese context (0.252) compared to its English capabilities (0.140). This phenomenon can be primarily attributed to the varying sizes of raw training text corpora across different languages. This observation aligns with the results reported in Chatbot Arena as of the time of writing, in which llama3.1-8b ranks 43rd for English but 55th for Chinese. The experimental results also show that `Markdown Awareness` is not affected by the task domain generally (see more in Appendix B.4).

### 6 Conclusion and Future Work

In this paper, we focus on an overlooked yet important metric — `Markdown Awareness`, which directly affects the readability and structure of the content in web chatbots (e.g., ChatGPT). To evaluate `Markdown Awareness` efficiently and effectively, we present MDEval, a novel LLM benchmark, which is a hybrid scorer with excellent interpretability by combining statistical-based and model-based methods. One of the main contributions of MDEval is the development of a ground-truth-free dataset which is tailored for this structure-oriented task with respect to Markdown quality. We also established a human alignment system from scratch to test the validity of our proposed method, and the experimental results demonstrate the superiority of MDEval. We believe this work is valuable to build better web chatbots while offering insights of enhancing the `Markdown Awareness` for LLMs. Furthermore, MDEval is highly scalable, allowing for easy expansion of datasets, incorporation of new models, and addition of human scoring records.

A possible future and ongoing work is to utilize MDEval to evaluate a greater variety of LLMs with larger datasets. We can even take account into LLMs' responses from a semantic perspective, and then evaluate the readability, coherence as well as structure in a holistic manner.

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

# A Ranking Methodology

## A.1 Design of D-Rule

Although a statistical scorer is often considered as less performant than model-based methods, in this work, we developed a reasonable method by heuristic rules, providing better performance than LLM-based implementations, as shown in Table 2. The main novelty of D-Rule is to introduce a delayed factor in the common weighted sum, and the intuition behind it lies in the fact that the repeated Markdown element has less effect in terms of Markdown Awareness. In D-Rule, we assign a weight $w_m$ to a Markdown element ($m$). Assume that there are $N$ elements in a response, and the decay factor is $\gamma$, then the score contributed by $m$ is:

$$s_m = \sum_{i=1}^{N} \gamma^{i-1} w_m$$

The final (unnormalized) score of a response is $\sum_{m \in \mathcal{M}} s_m$, where $\mathcal{M}$ is the set of all Markdown elements. In this way, D-Rule is able to alleviate the issue of preferring verbose responses, and the result is more accuracy. In this work, we use $\gamma = 0.5$. Note that there is no standard way to specify the weights, and in this paper, according to heuristic rules in terms of web content readability, the Markdown elements which have larger impact on content structure, including headings, code, math, list and bold, are assigned to 10, while other elements are assigned to 5.

## A.2 Average Ranking in RQ1

In Section 5.2, we propose a novel average ranking method which is inspired by GPA calculation from letter grades. To be specific, we re-assign a score on the well-studied 4.0 scale based on LLMs' ranking per task, and it could help smooth data by reducing the impact of extreme values.

**Table 3: Re-assigned Scores in Average Ranking**

| Ranking | Score |
|---------|-------|
| #1 | 4.0 |
| #2 | 3.7 |
| #3 | 3.3 |
| #4 | 3.0 |
| #5 | 2.7 |
| #6 | 2.3 |
| #7 | 2.0 |
| #8 | 1.7 |
| #9 | 1.3 |

## B Supplementary Experiments

## B.1 Human Alignment Evaluation System

Figure 9 and Figure 10 show the average and pairwise win rate, respectively.

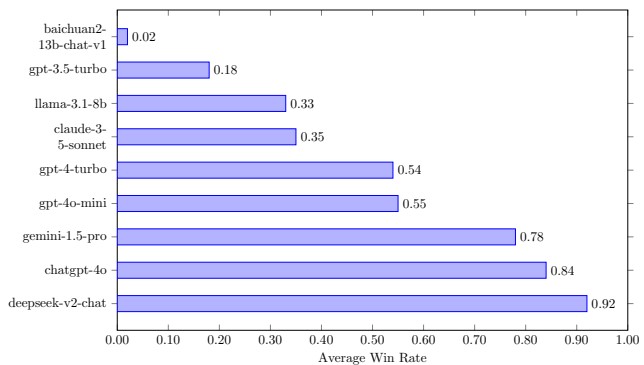

**Figure 9: Average win rate summary.**

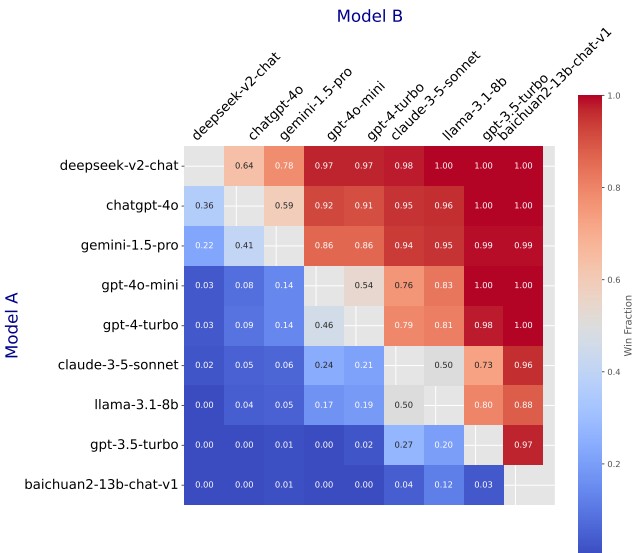

**Figure 10: Pairwise win rate summary.**

## B.2 RQ2

Different from Table 2, the accuracy and correlation of all methods with ties in this work are reported in Table 4.

**Table 4: Accuracy and Correlation with Ties**

| Method | Accuracy | Spearman | Pearson | Kendall |
|--------|----------|----------|---------|---------|
| MDEval | 63.4% | **0.791** | **0.844** | 0.670 |
| P-LLM | **65.4%** | 0.779 | 0.783 | **0.674** |
| R-LLM | 61.5% | 0.710 | 0.705 | 0.600 |
| D-Rule | 62.1% | 0.757 | 0.685 | 0.625 |

## B.3 RQ3

**Table 5: Pearson and Kendall's Tau Correlations by Category**

| Category | Pearson | Kendall |
|----------|---------|---------|
| Instruction Following | 0.567 | 0.317 |
| English | 0.576 | 0.411 |
| Chinese | 0.576 | 0.348 |
| Math | 0.568 | 0.243 |
| Coding | 0.571 | 0.344 |
| Hard Prompt (All) | 0.571 | 0.319 |
| Longer Query | 0.568 | 0.377 |

Table 5 shows both Pearson and Kendall correlations between Markdown Awareness and various capabilities of LLMs in the public leaderboard.

**Table 6: Average Scores Across Different Subjects**

| Model Name | Average | SUB1 | SUB2 | SUB3 | SUB4 | SUB5 | SUB6 | SUB7 | SUB8 | SUB9 | SUB10 |
|---|---|---|---|---|---|---|---|---|---|---|---|
| deepseek-v2-chat | 0.946 | 0.967 | 0.950 | 0.964 | 0.958 | **0.925** | **0.936** | **0.929** | 0.958 | 0.944 | 0.940 |
| gpt-4o | 0.865 | 0.913 | 0.871 | 0.910 | 0.925 | **0.822** | **0.832** | **0.839** | 0.878 | **0.848** | **0.824** |
| gpt-4o-mini | 0.830 | 0.891 | 0.827 | 0.854 | 0.823 | **0.788** | **0.808** | 0.825 | 0.872 | **0.813** | **0.807** |
| gemini-1.5-pro | 0.803 | 0.833 | **0.773** | **0.789** | 0.805 | **0.784** | **0.773** | 0.819 | 0.824 | 0.804 | 0.798 |
| gpt-4-turbo | 0.787 | 0.837 | 0.801 | 0.815 | **0.769** | **0.723** | **0.775** | 0.788 | 0.801 | 0.786 | **0.773** |
| llama-3.1-8b | 0.710 | **0.518** | **0.602** | **0.516** | 0.793 | 0.744 | 0.758 | 0.763 | 0.823 | 0.781 | 0.792 |
| calude-3.5-sonnet | 0.569 | 0.595 | 0.589 | 0.609 | 0.587 | **0.515** | **0.542** | 0.578 | 0.575 | 0.566 | **0.535** |
| gpt-3.5-turbo | 0.482 | 0.519 | 0.527 | 0.524 | 0.527 | **0.395** | **0.439** | 0.488 | 0.491 | 0.470 | **0.434** |
| baichuan2-13b-chat-v1 | 0.171 | 0.234 | 0.272 | 0.249 | 0.203 | **0.105** | **0.144** | **0.143** | **0.111** | 0.161 | **0.114** |

## B.4  QR5

Table 6 shows the results of average scores across different subjects in MDEval, maintaining the same order of subjects as listed in Section 4.1. For example, SUB7 means Education and Learning (Math, Programming, etc.). The score which is greater than or equal to average plus 0.01 is displayed using an underline, while the score which is smaller than or equal to average minus 0.01 is displayed in bold. Despite the variations, we can conclude that Markdown Awareness is not affected by the task domain.

Received 20 February 2007; revised 12 March 2009; accepted 5 June 2009

