# OpenReview forum: "MDEval: Evaluating and Enhancing Markdown Awareness in Large Language Models"
_ACM.org/TheWebConf/2025/Conference — WWW 2025 Poster_

### Official Review · Reviewer_hYx3 · 2024-11-24

**Novelty:** 4
**Technical Quality:** 3

**Review:**

The study focuses on an overlooked yet important metric — Markdown Awareness, which directly impacts the readability and structure of the content generated by these language models. The introduce MDEval, a comprehensive benchmark to assess Markdown Awareness for LLMs, by constructing a dataset with 20K instances covering 10 subjects in English and Chinese, which are still scarce in the literature. The study also propose the corresponding metrics to evaluate the Markdown Awareness of the model responses.

The most significant problem is that there is no connection with the track Search and retrieval-augmented AI. The author should determine which track is more relervant and clearly claim their connections.

**Questions:**

What is the connection to the Search and RAG?

**Reviewer Confidence:**

3: The reviewer is confident but not certain that the evaluation is correct

**Scope:**

1: The work is irrelevant to the Web

---

### Official Review · Reviewer_PEJi · 2024-11-28

**Novelty:** 4
**Technical Quality:** 4

**Review:**

This paper studies the problem related to readability of markdown language outputs generated by LLMs. They introduce MDEval benchmark for this task. This benchmark is reference free, which means that the generated outputs are not compared to a reference but are evaluated using an LLM. The prompts in this benchmark are extracted from Wikipedia. The evaluation approach first asks a capable LLM, such as GPT-4o to correct the markdown mistakes in the generated text and use this as the references. Then, the scoring happens based on the number of edits that is needed to convert generated output to the generated references. Finally, they perform an extensive set of experiments to study different aspects of the problem.

Pros:

1) It is a new task that can be really interesting to NLG community.
2) They performed an extensive set of experiments with this benchmark to show its effectiveness.

Cons:
1) While it is tried to explain the relevance of this work to search track, I am not convinced this is relevant to this track.
2) I have conceptual problem with this evaluation approach. A benchmark should be capable of evaluating any model, regardless of its capability. However, using a capable model to fix the formatting problem and using that as the reference might not be acceptable. This capable model itself can make incorrect generations in formatting which affects the evaluation.

**Questions:**

1) Why the judge, GPT-4o, does not get a score of 1? Is this because when it is asked to generate response it does not consider markdown format?

**Reviewer Confidence:**

3: The reviewer is confident but not certain that the evaluation is correct

**Scope:**

2: The connection to the Web is incidental, e.g., use of Web data or API

---

### Official Review · Reviewer_N2e1 · 2024-11-30

**Novelty:** 6
**Technical Quality:** 6

**Review:**

The paper introduces MDEval, a novel benchmark for evaluating and enhancing Markdown Awareness in large language models (LLMs). Markdown Awareness is a key metric for the readability and structure of responses in web chatbots, focusing on the use of Markdown elements (e.g., headings, bolding, lists) for structured outputs. The authors address three major challenges: insufficient datasets, so they propose a ground-truth-free dataset with 20.000 instances in English and Chinese across 10 subjects; metric validity, they introduce a structure-oriented scoring method using an intermediate rewriting phase to ensure explainability and reliability; metric quantity, where they compute Markdown Awareness through edit distance after converting responses into HTML. Results show that fine-tuning less performant models on this dataset enables them to achieve competitive Markdown Awareness compared to advanced proprietary models.

The paper is well-written and easy to follow. The concept and the objective is clear and the methodology well-executed.

**Minor issues**:
- The Saxon genitive is used incorrectly throughout the paper, it's applied to non-living entities
- "Calude-3.5-sonnet" is misspelt in several instances and should be corrected.
- Line 217 "phase" is mistakenly written as "phrase"
- Line 648/649 "GPT-40" is written instead of "GPT-4o".. Moreover, sometimes throughout the paper "GPT-4o" is mistakenly lowercase
- Sections from 5.2 to 5.6 (RQs) would benefit from the question in their names. As an example, when I was reading RQ4 I had forgotten what the research question was

**Questions:**

1. About the metric: The edit distance approach assumes structural alignment correlates with Markdown Awareness. Did you explore other metrics that consider semantic or visual impact?
2. How sensitive are the performance improvements to the size or quality of the fine-tuning dataset? Is there a point of diminishing returns? Moreover, did you assess whether, after fine-tuning, the model did (or did not) lose its chat capabilities?
3. Did you observe any specific patterns or common weaknesses in handling Markdown elements by the LLMs? For instance, were certain elements (e.g., tables, lists) more challenging or easier?

**Reviewer Confidence:**

4: The reviewer is certain that the evaluation is correct and very familiar with the relevant literature

**Scope:**

3: The work is somewhat relevant to the Web and to the track, and is of narrow interest to a sub-community

---

### Official Review · Reviewer_bz9m · 2024-12-02

**Novelty:** 5
**Technical Quality:** 5

**Review:**

Contributions:
1) Proposes the first benchmark, MDEval, to evaluate the quality of Markdown outputs generated by large language models, and introduces an innovative structure-oriented metric, namely “Markdown Awareness.”
2) Reports the Markdown Awareness performance of nine mainstream large language models based on this dataset.
3) Demonstrates that by fine-tuning underperforming open-source models on the constructed dataset, their Markdown Awareness performance can be improved to a level comparable to GPT-4.

Originality:
1) This paper focuses on an overlooked yet highly important metric—Markdown Awareness. It introduces a related benchmark and evaluation metrics, showcasing strong originality.

Quality:
1) The paper addresses the three major requirements of a benchmark—dataset construction, metric validity, and metric quantification—through logical reasoning, scientific implementation, and validation of their effectiveness, achieving an excellent benchmark design.
2) it evaluates the Markdown Awareness performance of nine mainstream large language models, producing high-quality results.
3) The comparison of baseline models and the related experimental evaluation metrics are rigorously demonstrated.
4) The validity of Markdown Awareness evaluation metrics is verified by comparing them against human consistency.

Clarity:
1) The challenges of the benchmark are addressed step by step, with clear and rigorous experimental designs that include adopting benchmarking methods and exploring factors influencing model performance.

Significance:
1) This work provides a new benchmark for evaluating the readability of large language models. However, there remains uncertainty regarding whether Markdown formatting itself impacts readability.

**Questions:**

1) Regarding the evaluation method of MDEval, where the original model output and the rewritten output are in Markdown format, you further convert this format to HTML for scoring. This raises some questions:
- What is your method of conversion?
- Does the content change during the conversion process? If the content itself changes, scoring based on the edit distance of HTML tags might have vulnerabilities.
- How can you ensure that the HTML scores reflect the Markdown format scores?
2) The paper proposes that using MDEval for fine-tuning can significantly enhance Markdown awareness. However, will the fine-tuned large model’s output accuracy be affected? The reason for this concern lies in the dataset creation process, where outputs with good formatting are considered acceptable prompts. How can we ensure that fine-tuning does not compromise the model’s ability to produce accurate outputs?
3) In the abstract, the authors highlight that the paper evaluates readability from the perspective of the structure of the output content. However, is Markdown truly the optimal format for this purpose? This requires further justification. Without evidence, the evaluation metric might lack strong practical value.
4) In an Intermediate Rewrite Phrase (C2) , the input is the original output of the large model, and the output is a revised Markdown-formatted version. In the corresponding prompt, it is stated:
	"Given the text below, rewrite it using Markdown format to make the output more structured and increase readability. Note that if possible, keep the content the same, just adjust the formatting."
Suppose the original output is already in Markdown format, and the revised output is also in Markdown, with only formatting differences. If both have the same readability, will the subsequent comparison and scoring yield identical scores? If the scores are not identical, could this lead to a misjudgment in the results?

**Reviewer Confidence:**

3: The reviewer is confident but not certain that the evaluation is correct

**Scope:**

4: The work is relevant to the Web and to the track, and is of broad interest to the community

---

### Official Review · Reviewer_4NUu · 2024-12-02

**Novelty:** 4
**Technical Quality:** 4

**Review:**

The paper proposes a new benchmark for evaluating Markdown Awareness in Large Language Models, which is a factor for readability and structuring web chatbot responses. The introduced benchmark MDEval integrates both model-based and statistical methods to assess Markdown output quality without explicit instruction in a zero-shot setting. They also introduce the concept of "Markdown Awareness," which they evaluate using a custom dataset with 20,000 instances covering ten subjects in English and Chinese.

Pros:
1. This paper introduces a factor for readability and structuring web chatbot responses, which could be important, especially in LLM evaluations.
2. This paper proposes a new and first benchmark namely MDEval for evaluating LLM's markdown quality.
3. The paper is written with good clarity and thus easy to follow.

Cons:
1. The primary concern lies in the scope of the paper, especially as a benchmark study for the WWW research track.
While the paper evaluates a range of diverse models, it lacks an in-depth analysis of why certain models perform better or worse in Markdown Awareness, which could provide valuable insights into the benchmark's implications and the evaluated LLMs.

**Questions:**

Please see the above weakness for details.

**Reviewer Confidence:**

3: The reviewer is confident but not certain that the evaluation is correct

**Scope:**

3: The work is somewhat relevant to the Web and to the track, and is of narrow interest to a sub-community